# Price of Parsimony: Complexity of Fourier Sparsity Testing

**Arijit Ghosh**
arijitiitkgpster@gmail.com
Indian Statistical Institute, Kolkata, India

**Manmatha Roy**
reach.manmatha@gmail.com
Indian Statistical Institute, Kolkata, India

## Abstract

A function $f : \mathbb{F}_2^n \to \mathbb{R}$ is said to be $s$-Fourier sparse if its Fourier expansion contains at most $s$ nonzero coefficients. In general, the existence of a sparse representation in the Fourier basis serves as a key enabler for the design of efficient learning algorithms. However, most existing techniques assume prior knowledge of the function's Fourier sparsity, with algorithmic parameters carefully tuned to this value. This motivates the following decision problem: given $s > 0$, determine whether a function is $s$-Fourier sparse.

In this work, we study the problem of tolerant testing of Fourier Sparsity for real-valued functions over $\mathbb{F}_2^n$, accessed via oracle queries. The goal is to decide whether a given function is close to being $s$-Fourier sparse or far from every $s$-Fourier sparse function. Our algorithm provides an estimator that, given oracle access to the function, estimates its distance to the nearest $s$-Fourier sparse function with query complexity $\widetilde{O}(s)$, for constant accuracy and confidence parameters.

A key structural ingredient in our analysis is a new spectral concentration result for real-valued functions over $\mathbb{F}_2^n$ when restricted to small-dimensional random affine subspaces. We further complement our upper bound with a matching lower bound of $\Omega(s)$, establishing that our tester is optimal up to logarithmic factors. The lower bound exploits spectral properties of a class of cryptographically hard functions, namely, the Maiorana–McFarland family, in a novel way.

## 1 Introduction

Sparsity is one of the most powerful ideas connecting modern machine learning and theoretical computer science. It captures the intuition that, even in high-dimensional settings, many natural functions or signals depend only on a small number of relevant components. This assumption underlies algorithms that are both sample- and time-efficient, forming the basis of techniques such as sparse linear regression, decision tree learning, and compressed sensing. Across these frameworks, the common principle is simple yet profound: if a function admits a sparse representation in a suitable basis, such as Fourier, wavelet, polynomial, or a learned dictionary, then learning and inference can be made dramatically more efficient.

A particularly elegant setting where sparsity plays a central role is that of real-valued functions over the Boolean hypercube $\mathbb{F}_2^n$. Fourier analysis provides a natural orthonormal basis for such functions. Every function $f : \mathbb{F}_2^n \to \mathbb{R}$ can be expressed as

$$f(x) = \sum_{\alpha \in \mathbb{F}_2^n} \widehat{f}(\alpha) \, (-1)^{\langle x, \alpha \rangle},$$

where $\widehat{f}(\alpha)$ denotes the Fourier coefficient at frequency $\alpha$. The *Fourier sparsity* of $f$, the number of nonzero coefficients in this expansion quantifies how succinctly the function can be represented in the Fourier basis.

39th Conference on Neural Information Processing Systems (NeurIPS 2025).

Fourier sparsity is a recurring theme across many areas of theoretical computer science. In complexity theory, it lies at the heart of problems involving communication complexity and parity decision trees for Boolean functions $f : \mathbb{F}_2^n \to \{+1, -1\}$ [32, 33, 29, 27, 10]. In learning theory, it has become a central tool for designing algorithms that efficiently learn functions with low-degree or low-support Fourier spectra. Many natural Boolean functions exhibit this property: graph and hypergraph cut functions, as well as decision trees of bounded depth, are inherently Fourier sparse because their spectra are concentrated on low-degree coefficients [28, 22]. For example, the cut function of a graph corresponds to a degree-2 polynomial in the Fourier basis, while a degree-$d$ hypergraph cut function has degree at most $d$. Similarly, a Boolean decision tree of depth $d$ has its spectrum supported entirely on coefficients of degree at most $d$.

Beyond these classical examples, Fourier-sparse models have found renewed relevance in modern machine learning. They appear in settings such as neural network hyperparameter optimization [20] and the learning of structured set functions [4]. The impact of Fourier sparsity extends even further, into cryptography, where the celebrated Goldreich–Levin theorem [13] established a deep connection between identifying large Fourier coefficients and constructing hardcore predicates for one-way functions.

Algorithmically, two main approaches have emerged for learning or recovering Fourier-sparse functions: *Sparse Hadamard Transform* methods [17, 25, 21] and *Compressed Sensing* techniques [24, 19]. Both families of algorithms, however, rely critically on prior knowledge of the function's sparsity level. This partcular gap motivates our work, which focuses on efficiently estimating the Fourier sparsity, up to a desired approximation in $\ell_2$ distance. Such an estimator can serve as a useful preprocessing step in learning pipelines, both for verifying whether sparsity-based assumptions hold and for guiding the initialization of sparsity parameters in downstream algorithms.

To formalize this setting, we first introduce some basic definitions. We measure distances between functions using the squared $\ell_2$-norm:

$$\mathrm{dist}_2^2(f, g) := \|f - g\|_2^2 = \frac{1}{2^n} \sum_{x \in \mathbb{F}_2^n} (f(x) - g(x))^2.$$

For a function $f$ and a class of functions $\mathcal{P}$, the distance of $f$ from $\mathcal{P}$ is defined as

$$\mathrm{dist}_2^2(f, \mathcal{P}) := \min_{g \in \mathcal{P}} \|f - g\|_2^2.$$

We also recall the standard definition of the $\ell_2$-norm. For any function $f : \mathbb{F}_2^n \to \mathbb{R}$,

$$\|f\|_2^2 := \frac{1}{2^n} \sum_{x \in \mathbb{F}_2^n} f(x)^2.$$

Let $\mathcal{F}_s$ denote the class of all *s-Fourier sparse functions*, that is, functions $f : \mathbb{F}_2^n \to \mathbb{R}$ whose Fourier spectrum has at most $s$ nonzero coefficients. We are interested in determining how close a given function $f$ is to this class.

**Problem 1.1.** *Given query access to a function $f : \mathbb{F}_2^n \to \mathbb{R}$ with $\|f\|_2^2 = 1$, and parameters $s > 0$, $\epsilon \in (0, 1]$, and $\delta \in [0, 1]$, the task is to design a randomized algorithm that distinguishes between the following two cases:*

- *(Close): There exists $g \in \mathcal{F}_s$ such that $\|f - g\|_2^2 \leq \delta$.*

- *(Far): For every $g \in \mathcal{F}_s$, $\|f - g\|_2^2 \geq \delta + \epsilon$.*

*The goal is to construct such an algorithm using as few queries to $f$ as possible, while ensuring that it distinguishes the two cases with high probability.*

Our main contribution in this paper is the design of a simple, nonadaptive and almost optimal query algorithm for testing Fourier sparsity.

**Theorem 1.2.** *Let $s > 0$, $\epsilon \in (0, 1]$, and $\delta \in [0, 1]$. Let $f : \mathbb{F}_2^n \to \mathbb{R}$ be an unknown function with $\|f\|_2 = 1$, accessible only via query access. Then, there exists a* nonadaptive *algorithm that, with success probability at least $2/3$, distinguishes between the following two cases:*

- *$f$ is $\delta$-close to being $s$-Fourier sparse,*

- *f is $(\delta + \epsilon)$-far from every $s$-Fourier sparse function,*

using at most $\widetilde{O}(s/\epsilon^2)$ queries to $f$, where the $\widetilde{O}(\cdot)$ notation hides factors polynomial in $\log s$ and $\log(1/\epsilon)$.

Theorem 1.2 is proved in Section 3. Although stated under the assumption that the function has unit $\ell_2$-norm, the result extends to any nonzero function $f : \mathbb{F}_2^n \to \mathbb{R}$ via normalization.

We also show that the query complexity of our algorithm is tight up to logarithmic factors by proving a matching lower bound.

**Theorem 1.3.** *Let $s > 0$. Any randomized algorithm that decides whether a function $f : \mathbb{F}_2^n \to \{-1, +1\}$, is $s$-Fourier sparse or $(1/4)$-far from every $s$-Fourier sparse function over $\mathbb{F}_2^n$, must make $\Omega(s)$ queries to $f$ to succeed with probability at least $2/3$.*

The reader may note that any Boolean function $f : \mathbb{F}_2^n \to \{-1, +1\}$ satisfies $\|f\|_2^2 = 1$. The proof of Theorem 1.3 is presented in Section 4.

## 1.1 Related works

Testing Fourier sparsity was first studied by Gopalan et al. [16], who focused on Boolean functions and used the *Hamming distance* as a measure of proximity. (The Hamming distance between two functions $f, g : \mathbb{F}_2^n \to \{0, 1\}$ is the fraction of inputs on which they differ.) Their algorithm has query complexity $O(s^{14})$, which quickly becomes impractical for even moderately large values of $s$. Similarly, the regularity framework of Hatami and Lovett [18] provides a general-purpose, black-box reduction for testing Fourier sparsity under Hamming distance, but this approach suffers from a query complexity that grows as a tower function in $s$.

In the real-valued setting, Yaroslavtsev and Zhou [31] considered testing Fourier sparsity with respect to the squared $\ell_2$-distance. They designed an algorithm with query complexity $\widetilde{O}(s/\epsilon^4)$ and established a lower bound of $\Omega(\sqrt{s})$. In comparison, our algorithm improves the dependence on the proximity parameter $\epsilon$, requiring only $\widetilde{O}(s/\epsilon^2)$ queries, and is conceptually simpler. We further establish a nearly tight lower bound of $\Omega(s)$, quadratically improving the current state of art [31], showing that our algorithm is optimal up to logarithmic factors.

It is important to note that testing Fourier sparsity in the *random example* model is significantly more challenging. As shown in [14], even for linearity testing, where the target functions are 1-Fourier sparse, it is not known how to design a tester whose sample complexity is independent of the ambient dimension $n$. In contrast, in the *query access* model, linearity testing can be performed efficiently using the well-known 3-query BLR test. Our work aims to estimate the Fourier sparsity level of a function in a way that depends only on the sparsity $s$ and the proximity parameter $\epsilon$, while remaining independent of the ambient dimension $n$.

## 1.2 Proof Sketch of Theorem 1.2.

The design of our tester is inspired by the well-established framework for testing hereditary graph properties. A canonical tester for such a property $P$ samples a small random subset of vertices, queries all induced edges, and checks whether the resulting subgraph satisfies $P$. For hereditary properties, those preserved under taking induced subgraphs, this approach guarantees only a modest (quadratic) overhead in query complexity [2, 15, 1]. Indeed, for several natural properties, such as bipartiteness, the canonical tester achieves optimal performance up to constant factors.

A similar idea has been successfully adapted to Boolean functions, particularly for testing affine-invariant properties [8, 6]. In this setting, the canonical approach restricts the function to random low-dimensional affine subspaces and tests the property on these restrictions. While this strategy enjoys strong generality and theoretical support, e.g., via regularity-like lemmas, it often suffers from impractical query complexity, including tower-type dependencies [18]. Nonetheless, specialized testers exploiting finer structural properties have been developed for specific cases, such as low algebraic degree [3] and odd-cycle-freeness [7]. Surprisingly, despite being a natural and central affine-invariant property, Fourier sparsity has largely resisted similar progress.

Prior works in Fourier sparsity testing typically project the Fourier spectrum into randomly chosen cosets of sufficiently large codimension, a process commonly referred to as *Fourier hashing*, which

was originally introduced in [11]. Analytical or combinatorial tools are then applied to extract sparsity information. For example, Gopalan et al. [16] presents a granularity theorem for Fourier-sparse functions, showing that individual coefficients cannot be too small and reducing the problem to counting large-weight cosets. Similarly, Yaroslavtsev et al. [31] certain concentration of the $\ell_2$-norm of heavy buckets to design their tester.

In contrast, our approach analyzes the function restricted to a randomly chosen subspace. We approximately recover the Fourier spectrum of this restricted function and use it to infer the sparsity of the original function. A new structural relationship between the Fourier coefficients of the restricted and original functions shows that, under suitable subspace choices, their magnitudes closely match. This relationship is central to our analysis. Instead of explicitly defining a hashing process, restricting the function to a subspace implicitly induces a hashing, allowing us to derive a concentration bound in terms of the $\ell_1$-norm of bucketed Fourier coefficients, which constitutes our main technical contribution.

### 1.3 Proof Sketch of Theorem 1.3.

We prove a lower bound for testing Fourier sparsity via a reduction from randomized communication complexity, following the approach introduced by Blais, Brody, and Matulef [9]. Our reduction builds on the structure of Maiorana–McFarland functions and their connection to the Approximate Matrix Rank problem. Maiorana–McFarland functions are widely used in theoretical computer science, especially for circuit lower bounds and structural analysis of Boolean functions. They also play a central role in symmetric-key cryptography, thanks to their spectral properties that support strong confusion and diffusion.

Consider a communication problem where Alice and Bob receive matrices $A, B \in \mathbb{F}_2^{m \times n}$, and their goal is to determine whether the sum $C = A + B$ has rank at least $R$, or at most $cR$, for some fixed constant $c < 1$. We encode this instance into the Fourier domain by composing Maiorana–McFarland functions with linear transformations derived from the input matrices. A central property of this construction is that the Fourier sparsity of the resulting function is closely tied to the rank of the matrix $C$. Thus, distinguishing high-rank from low-rank instances in the matrix problem reduces to distinguishing functions that are close to being Fourier sparse from those that are far.

To complete the reduction, we use a result of Sherstov and Storozhenko [26], which shows that any randomized protocol for the Approximate Matrix Rank problem must communicate at least $\Omega(R^2)$ bits. Since our reduction incurs only a constant overhead, we conclude that any nonadaptive algorithm for testing Fourier sparsity must make $\Omega(R^2)$ queries in the worst case. This matches our upper bound up to logarithmic factors and establishes the optimality of our tester.

## 2 Background

Any function $f : \mathbb{F}_2^n \to \mathbb{R}$ can be uniquely expressed as

$$f(x) = \sum_{\alpha \in \mathbb{F}_2^n} \widehat{f}(\alpha)\chi_\alpha(x),$$

where $\chi_\alpha(x) = (-1)^{\langle \alpha, x \rangle}$ and $\widehat{f}(\alpha) = \mathbb{E}_x[f(x)\chi_\alpha(x)]$. The quantity $\widehat{f}(\alpha)^2$ denotes the Fourier weight on $\alpha$, and the collection $\{\widehat{f}(\alpha)\}$ is the Fourier spectrum of $f$. We use the following standard facts:

- **Parseval's identity:** $\|f\|_2^2 = \sum_\alpha \widehat{f}(\alpha)^2$.
- **Plancherel's theorem:** $\langle f, g \rangle = \sum_\alpha \widehat{f}(\alpha)\widehat{g}(\alpha)$.
- **Character multiplication:** $\chi_{\alpha+\beta} = \chi_\alpha \chi_\beta$.
- **Poisson summation:** For any subspace $H \subseteq \mathbb{F}_2^n$, we have

$$\sum_{x \in H} \chi_\alpha(x) = \begin{cases} |H|, & \alpha \in H^\perp, \\ 0, & \text{otherwise.} \end{cases}$$

For our lower bound theorem, we will require the following definitions and results from communication complexity.

In the randomized communication model, Alice and Bob compute a function $f : X \times Y \to \{0, 1\}$ using shared randomness. The randomized communication complexity $R_{1/3}(f)$ is the minimum number of bits exchanged to compute $f(x, y)$ correctly with probability at least $2/3$.

In the *Approximate Matrix Rank* problem, Alice holds $A \in \mathbb{F}_2^{r \times r}$ and Bob holds $B \in \mathbb{F}_2^{r \times r}$; they must distinguish whether $\mathrm{rank}(A + B) = r$ or $\frac{r}{4}$. The following lower bound is known [26, Theorem 1.1]:

$$R_{1/3}\Big(\mathrm{RANK}_{\mathbb{F}_2, r, r}^{r, \frac{r}{4}}\Big) = \Omega(r^2).$$

## 3 Improved upper bound for testing Fourier sparsity

Our analysis centers on restricting the function $f$ to random affine subspaces of $\mathbb{F}_2^n$. We study how the individual Fourier coefficients behave under such restrictions, comparing it to that of the original function. Table 1 summarises the notations used in this section.

### 3.1 Fourier analysis under affine restrictions

We consider a function $f : \mathbb{F}_2^n \to \mathbb{R}$. Let $H \subseteq \mathbb{F}_2^n$ be a subspace and $\alpha \in \mathbb{F}_2^n$. Define the *restricted function* $f_A : H \to \mathbb{R}$ by
$$f_A(x) = f(x + \alpha), \quad \forall x \in H.$$
We will briefly recall some standard facts about the Fourier spectrum of $f_A$. Let $H^\perp \subseteq \mathbb{F}_2^n$ be the *annihilator* (see Table 1) of $H$, that is, the set of vectors orthogonal to every element of $H$, and let $W \subseteq \mathbb{F}_2^n$ be a *complementary subspace* to $H^\perp$, so that
$$\mathbb{F}_2^n = W \oplus H^\perp \text{ and } W \cap H^\perp = \{0^n\}.$$
The Fourier coefficients of $f_A$ are naturally indexed by $\gamma \in W$, and the Fourier expansion of $f_A$ is
$$f_A(x) = \sum_{\gamma \in W} \widehat{f_A}(\gamma) \, \chi_\gamma(x), \quad x \in H,$$

where
$$\widehat{f_A}(\gamma) = \sum_{\beta \in \gamma + H^\perp} \widehat{f}(\beta) \, \chi_\beta(\alpha),$$

and $\widehat{f}(\beta)$ are the Fourier coefficients of $f$ on $\mathbb{F}_2^n$. Observe that there is another way to write the Fourier expansion of $f_A$ in the following way:
$$f_A(x) = \sum_{\widetilde{\beta} \in \mathbb{F}_2^n / H^\perp} \widehat{f_A}(\widetilde{\beta}) \, \chi_\gamma(x), \quad x \in H,$$

where
$$\widehat{f_A}(\widetilde{\beta}) = \sum_{\beta \in \widetilde{\beta}} \widehat{f}(\beta) \, \chi_\beta(\alpha).$$

Recall that $\widetilde{\beta}$ is a coset of $H^\perp$ and therefore a subset of $\mathbb{F}_2^n$.

**Remark 3.1.** *By abuse of notation, for a $\gamma \in \mathbb{F}_2^n$, $\widehat{f_A}(\gamma)$ denotes the Fourier coefficient $\widehat{f_A}(\widetilde{\gamma})$ corresponding to the coset $\widetilde{\gamma} = \gamma + H^\perp$ of $H^\perp$ containing $\gamma$.*

The following identities will be used by our tester, given in Section 3.3, for estimating Fourier coefficients of functions restricted to a affine subspace.

**Theorem 3.2.** *Let $A = H + \alpha$, where $H$ be a subspace of $\mathbb{F}_2^n$ and $\alpha \in \mathbb{F}_2^n$.*

*(a) For all $\gamma$ in the complementary subspace $W$ of $H^\perp$, we have*

$$\widehat{f_A}(\gamma) = \mathop{\mathbb{E}}_{x \in H} [f_A \chi_\gamma] := \frac{1}{|H|} \sum_{x \in H} f_A(x) \chi_\gamma(x).$$

*(b) Define $\|f_A\|_2^2 := \frac{1}{|H|} \sum_{x \in H} f_A(x)^2$. We have*

$$\|f_A\|_2^2 = \sum_{\gamma \in W} f_A(\gamma)^2.$$

Table 1: of Notation

| Notation | Meaning |
|---|---|
| $\langle \alpha, \beta \rangle$ | Inner product of $\alpha, \beta \in \mathbb{F}_2^n$, defined as $\langle \alpha, \beta \rangle := \sum_{i=1}^{n} \alpha_i \beta_i$ (over $\mathbb{F}_2$). |
| $\mathbb{E}_{x \in \mathbb{F}_2^n}[f]$ | Expectation of $f : \mathbb{F}_2^n \to \mathbb{R}$, $\mathbb{E}_{x \in \mathbb{F}_2^n}[f] := \frac{1}{2^n} \sum_{x \in \mathbb{F}_2^n} f(x)$. |
| $\langle f, g \rangle$ | Inner product of $f, g : \mathbb{F}_2^n \to \mathbb{R}$, $\langle f, g \rangle := \mathbb{E}_{x \in \mathbb{F}_2^n}[f(x)g(x)] = \frac{1}{2^n} \sum_{x \in \mathbb{F}_2^n} f(x)g(x)$. |
| $H$ | A (randomly chosen) linear subspace of $\mathbb{F}_2^n$. |
| $H^\perp$ | Given a subspace $H \subseteq \mathbb{F}_2^n$, $H^\perp$ denotes the *annihilator* of $H$, that is, $H^\perp := \{x \in \mathbb{F}_2^n : \langle x, h \rangle, \ h \in H\}$. |
| $A$ | An affine subspace of the form $\alpha + H$, where $\alpha \in \mathbb{F}_2^n$. |
| $f_A$ | Restriction of $f$ to the affine subspace $A$. |
| $\widehat{f_A}(\gamma)$ | Fourier coefficient of $f_A$ at $\gamma \in H$. |
| $\gamma^*$ | Element $\gamma^* := \arg\max_{\beta \in \gamma + H^\perp} |\widehat{f}(\beta)|$. |

## 3.2 Concentration of the Fourier spectrum under random affine restrictions

In Algorithm 1, the function $f$ is restricted to a uniformly random affine subspace $A = \alpha + H$. This affine subspace is constructed as follows: we first select $t$ vectors $h_1, h_2, \ldots, h_t$ independently and uniformly at random from $\mathbb{F}_2^n$, and define the linear subspace

$$H = \text{span}\{h_1, h_2, \ldots, h_t\}.$$

Next, to introduce a random shift of $H$, we choose $\alpha \in \mathbb{F}_2^n$ uniformly at random and independently of $H$, and define the affine subspace as

$$A = \alpha + H.$$

Observe that the collection of cosets $\{\gamma + H^\perp : \gamma \in H\}$ forms a partition of the space $\mathbb{F}_2^n$. Interestingly, the following lemma shows that this random coset partition behaves like a pairwise independent hash family over $\mathbb{F}_2^n$.

**Lemma 3.3** (Coset Hashing via Random Subspaces). *Let $H \subseteq \mathbb{F}_2^n$ be a uniformly random linear subspace constructed by taking span of $t$ random vectors from $\mathbb{F}_2^n$ sampled independently and uniformly from $\mathbb{F}_2^n$. Then the following hold:*

*1. For any distinct $\alpha, \beta \in \mathbb{F}_2^n$,*

$$\Pr_H \left[ \alpha, \beta \text{ lie in the same coset of } H^\perp \right] = \Pr_H [ \alpha - \beta \in H^\perp ] = 2^{-t}.$$

*2. For any subset $S \subseteq \mathbb{F}_2^n$ with $|S| \leq s$, if $t \geq 2 \log s + \log 100$, then*

$$\Pr_H \left[ \text{all elements of } S \text{ lie in distinct cosets of } H^\perp \right] \geq 0.99.$$

This lemma is a slight restatement of Proposition 3 from [16], modified to suit the specific needs of our proof. While the full proof is deferred to the appendix, we will assume it for the time being. We now show that, for a uniform random choice of affine subspace $A$, the magnitude of the Fourier coefficients of the restricted function $\widehat{f_A}(\gamma)$ is tightly concentrated around the magnitude of the largest Fourier coefficient of $f$ within the coset $\gamma + H^\perp$. Specifically, we define the leader of the coset as $\gamma^* = \arg\max_{\beta \in \gamma + H^\perp} |\widehat{f}(\beta)|$. From this point on, we refer to $\gamma^*$ as the leader of the coset. We now formally state the following concentration result.

**Lemma 3.4.** *Let $A = \alpha + H$ be a random affine subspace of $\mathbb{F}_2^n$, where $H$ is obtained as the span of $t$ vectors sampled independently and uniformly from $\mathbb{F}_2^n$, and $\alpha \in \mathbb{F}_2^n$ is an independent uniformly*

*random shift. Consider a function* $f : \mathbb{F}_2^n \to \mathbb{R}$ *with* $\|f\|_2^2 = 1$. *If* $t \geq \log \frac{1}{\eta^4}$, *then for every* $\gamma \in H$ *and every* $\tau > 0$,

$$\Pr\left[\left|\widehat{f_A}(\gamma) - \widehat{f}(\gamma)\chi_\gamma(\alpha)\right| > \eta + \tau\right] \leq \frac{\eta^4}{\tau^2}. \tag{1}$$

*Proof.* Fix $\gamma \in \mathbb{F}_2^n$. Recall, from Section 3.1, the Fourier coefficients of the restriction $f_A$ satisfies

$$\widehat{f_A}(\gamma) = \sum_{\beta \in \gamma + H^\perp} \widehat{f}(\beta)\chi_\beta(\alpha).$$

Consider the following random variable:

$$X := \widehat{f_A}(\gamma) - \widehat{f}(\gamma)\chi_\gamma(\alpha) = \sum_{\substack{\beta \in \gamma + H^\perp \\ \beta \neq \gamma}} \widehat{f}(\beta)\chi_\beta(\alpha),$$

and let $Y := |X|$. All probabilities/expectations below are over the joint randomness of $H$ and $\alpha$; for fixed $H$ we write $\mathbb{E}_\alpha[\cdot \mid H]$.

**First moment bound.** For fixed $H$, using linearity of expectation, we have

$$\mathbb{E}_\alpha[X \mid H] = \mathbb{E}_\alpha\left[\sum_{\substack{\beta \in \gamma + H^\perp \\ \beta \neq \gamma}} \widehat{f}(\beta)\chi_\beta(\alpha)\right] = \sum_{\substack{\beta \in \gamma + H^\perp \\ \beta \neq \gamma}} \widehat{f}(\beta)\,\mathbb{E}_\alpha[\chi_\beta(\alpha)].$$

Now, observe that $\mathbb{E}_\alpha[\chi_\beta(\alpha)] = 0$ for all $\beta \neq 0$ and equals 1 for $\beta = 0$. Therefore,

$$\mathbb{E}_\alpha[X \mid H] = \begin{cases} \widehat{f}(0), & \text{if } 0 \in \gamma + H^\perp \text{ and } \gamma \neq 0 \\ 0, & \text{otherwise} \end{cases}$$

This implies that the expression of $\mathbb{E}_\alpha[X \mid H]$ can be rewritten in the following form:

$$\mathbb{E}_\alpha[X \mid H] = \widehat{f}(0)\mathbb{1}_{\{0 \in \gamma + H^\perp \text{ and } \gamma \neq 0\}}.$$

Using the fact that $\mathbb{E}_H[\mathbb{E}_\alpha[X \mid H]] = \mathbb{E}_{H,\alpha}[X]$, we get

$$\mathbb{E}_{H,\alpha}[X] = \widehat{f}(0) \cdot \Pr_H\left[0 \in \gamma + H^\perp \text{ and } \gamma \neq 0\right]$$

Using the fact that $\Pr_H\left[0 \in \gamma + H^\perp \text{ and } \gamma \neq 0\right] \leq \Pr_H\left[0 \in \gamma + H^\perp\right]$, we get

$$|\mathbb{E}_{H,\alpha}[X]| = |\widehat{f}(0)| \cdot \Pr_H\left[0 \in \gamma + H^\perp \text{ and } \gamma \neq 0\right] \leq |\widehat{f}(0)| \cdot \Pr_H\left[0 \in \gamma + H^\perp\right]$$

Observe that

$$\Pr_H\left[0 \in \gamma + H^\perp\right] = \Pr_H[\gamma \in H^\perp] = 2^{-t} \leq \eta^4.$$

The last inequality follows from the fact that $t \geq \log(1/\eta^4)$. Using the above bound on $\Pr_H[0 \in \gamma + H^\perp]$ and the fact that $|\widehat{f}(0)| \leq 1$ (from Parseval identity), we get

$$|\mathbb{E}_{H,\alpha}[X]| \leq \eta^4 \leq \eta.$$

**Second moment bound.** Observe that

$$X^2 = \sum_{\substack{\beta,\beta' \in \gamma + H^\perp \\ \beta,\beta' \neq \gamma}} \widehat{f}(\beta)\widehat{f}(\beta')\,\chi_\beta(\alpha)\chi_{\beta'}(\alpha) = \sum_{\substack{\beta,\beta' \in \gamma + H^\perp \\ \beta,\beta' \neq \gamma}} \widehat{f}(\beta)\widehat{f}(\beta')\,\chi_{\beta+\beta'}(\alpha),$$

since $\chi_\beta\chi_{\beta'} = \chi_{\beta+\beta'}$.

Taking expectation over $\alpha$ uniformly from $\mathbb{F}_2^n$, we use

$$\mathbb{E}_\alpha[\chi_\delta(\alpha)] = \begin{cases} 1 & \text{if } \delta = 0, \\ 0 & \text{if } \delta \neq 0. \end{cases}$$

Hence, only terms with $\beta + \beta' = 0$ contribute. Over $\mathbb{F}_2^n$ this means $\beta' = \beta$, so we obtain

$$\mathbb{E}_\alpha[X^2 \mid H] = \sum_{\substack{\beta,\beta' \in \gamma+H^\perp \\ \beta,\beta' \neq \gamma}} \widehat{f}(\beta)\widehat{f}(\beta') \, \mathbb{E}_\alpha[\chi_{\beta+\beta'}(\alpha)] = \sum_{\substack{\beta \in \gamma+H^\perp \\ \beta \neq \gamma}} \widehat{f}(\beta)^2.$$

Like in the case with "first moment calculation", we need to rewrite the expression of $\mathbb{E}_\alpha[X^2 \mid H]$ in terms of indicator random variables:

$$\mathbb{E}_\alpha[X^2 \mid H] = \sum_{\beta \in \mathbb{F}_2^n} \widehat{f}(\beta) \mathbb{1}_{\{\beta \in \gamma+H^\perp \text{ and } \beta \neq \gamma\}}$$

Taking expectation over $H$, we get

$$\mathbb{E}_{H,\alpha}[X^2] \leq \sum_{\beta \in \mathbb{F}_2^n} \widehat{f}(\beta)^2 \cdot \Pr_H[\beta \in \gamma+H^\perp].$$

For any fixed $\beta$, $\Pr_H[\beta \in \gamma + H^\perp] = \Pr_H[\beta + \gamma \in H^\perp] = 2^{-t}$. By Parseval $\sum_\beta \widehat{f}(\beta)^2 = 1$, so

$$\mathbb{E}_{H,\alpha}[X^2] \leq 2^{-t} \leq \eta^4.$$

Thus $\mathrm{Var}_{H,\alpha}(X) \leq \eta^4$.

**Applying Chebyshev inequality.** Since $|\mathbb{E}_{H,\alpha}[X]| \leq \eta$, the event $\{|X| > \eta + \tau\}$ implies $\{|X - \mathbb{E}_{H,\alpha}[X]| > \tau\}$. Hence

$$\Pr_{H,\alpha}\left[|X| > \eta + \tau\right] \leq \Pr_{H,\alpha}\left[|X - \mathbb{E}_{H,\alpha}[X]| > \tau\right] \leq \frac{\mathrm{Var}_{H,\alpha}(X)}{\tau^2} \leq \frac{\eta^4}{\tau^2}.$$

Since $X = \widehat{f_A}(\gamma) - \widehat{f}(\gamma)\chi_\gamma(\alpha)$, this proves the theorem. $\qquad\square$

---

**Algorithm 1:** FOURIER-SPARSITY-TESTER

---

**Input:** Tolerance parameter $\delta \geq 0$, proximity parameter $\epsilon > 0$, sparsity parameter $s$, and oracle access to a function $f : \mathbb{F}_2^n \to \mathbb{R}$ satisfying $\|f\|_2 = 1$

**Output:** YES if $\mathrm{dist}_2^2(f, \mathcal{F}_s) \leq \delta$, and NO if $\mathrm{dist}_2^2(f, \mathcal{F}_s) \geq \delta + \epsilon$

**Procedure:**
1: Choose a random affine subspace $A = \alpha + H \subseteq \mathbb{F}_2^n$, with $\dim(H) = \Theta\left(\log\left(s^2/\epsilon^2\right)\right)$

2: Let $f_A$ denote the restriction of $f$ to $A$, and compute an estimate $\widetilde{\mu}$ of the sum of squares of top $s$ Fourier coefficients of $f_A$ in terms of their absolute values

3: If $\widetilde{\mu} \geq 1 - (\delta + \epsilon/2)$ then YES, else NO.

---

## 3.3 Proof of Theorem 1.2

In this section, we present the algorithm and its analysis, completing the proof of Theorem 1.2.

**A structural characterization.** We begin with a simple observation that reduces testing Fourier sparsity to estimating spectral mass.

**Lemma 3.5** (Structural observation). *Let* $f : \mathbb{F}_2^n \to \mathbb{R}$ *with* $\|f\|_2 = 1$. *Then* $\mathrm{dist}_2^2(f, \mathcal{F}_s) = 1 - \max_{T \subseteq \mathbb{F}_2^n : |T| \leq s} \sum_{\beta \in T} \widehat{f}(\beta)^2$.

**The algorithm.** Algorithm 1 (FOURIER-SPARSITY-TESTER) selects a random affine subspace $A = \alpha + H \subseteq \mathbb{F}_2^n$ of dimension $t = \Theta\left(\log\left(s^2/\epsilon^2\right)\right)$, restricts the function to $A$, and estimates the Fourier coefficients of the restricted function $f_A$. It then computes the sum of squares of the largest $s$ estimated coefficients and compares this quantity to the threshold $1 - (\delta + \epsilon/2)$.

Since $|A| = 2^t$, the overall query complexity depends only on $s$ and $\epsilon$, and is independent of $n$.

**Correctness with exact restricted coefficients.** We first analyze the tester assuming exact access to the Fourier coefficients of $f_A$. Using the concentration of Fourier spectrum under random restrictions (Lemmas 3.3 and 3.4), we show that the sum of the squares of the top $s$ Fourier coefficients (in terms of their absolute values) of $f_A$ approximates from that of $f$ by an additive error $\approx \epsilon/4$.

If $f$ is $\delta$-close to $\mathcal{F}_s$, Lemma 3.5 implies that the top $s$ Fourier coefficients of $f$ carry mass at least $1 - \delta$, and hence the same holds for $f_A$ up to an additive error $\approx \epsilon/4$. Conversely, if $f$ is $(\delta + \epsilon)$-far from $\mathcal{F}_s$, then every set of $s$ Fourier coefficients of $f_A$ has total squared mass at most $1 - (\delta + \epsilon/2)$. Thus, the tester correctly distinguishes the two cases under exact Fourier access to $f_A$.

**Working with the estimates of the Fourier coefficients of $f_A$.** We first estimate the Fourier coefficients of $f_A$ from oracle access to $f$. We refer the reader to Section 3.1 for a brief introduction to the Fourier coefficients of the restricted function $f_A$. For each $\gamma \in W$, where $W$ is the complementary subspace of $H^\perp$ (see Table 1), we have

$$\widehat{f_A}(\gamma) = \mathbb{E}_{x \in H}\left[ f_A(x) \chi_\gamma(x) \right].$$

Using median-of-means technique [30, Exercise 2.2.9], we obtain the following.

**Lemma 3.6** (Fourier estimation on a subspace). *There exists a nonadaptive estimator that, using* $\widetilde{O}\left(s/\epsilon^2\right)$ *queries to $f$, produces estimates* $\left\{ \widetilde{\widehat{f_A}}(\gamma) \right\}_{\gamma \in W}$ *such that*

$$\max_{\gamma \in W} \left| \widetilde{\widehat{f_A}}(\gamma) - \widehat{f_A}(\gamma) \right| \leq \frac{\epsilon}{1000\sqrt{s}}$$

*with probability at least* $0.99$. *Note that $W$ denotes the complementary subspace of $H^\perp$.*

Hence, the sum of squares of the top $s$ estimated coefficients is within $\epsilon/50$ of the true value and does not affect the tester's decision.

**Completing the proof.** Combining the above arguments, Algorithm 1 distinguishes functions that are $\delta$-close to being $s$-Fourier sparse from those that are $(\delta + \epsilon)$-far using $\widetilde{O}(s/\epsilon^2)$ nonadaptive queries, with success probability at least $2/3$. This completes the proof of Theorem 1.2.

**Remark.** Complete proofs and technical details for this section appear in the full version [12].

## 4 Improved lower bound for testing Fourier sparsity

In this section, we prove Theorem 1.3. We begin by reviewing Maiorana–McFarland functions and the key properties required for the proof.

### 4.1 Spectral structure of Maiorana-McFarland functions

Variants of Maiorana–McFarland functions have found widespread use in theoretical computer science, particularly in proving circuit lower bounds and studying structural properties of Boolean functions relevant to complexity theory. They also play an important role in symmetric-key cryptography, especially in the design of stream ciphers. We now define them in their most general form.

Given positive integers $n$ and $r$ with $r \leq n$, the Maiorana–McFarland family $\mathrm{MM}_{r,n}$ [23] consists of $n$-variable Boolean functions $f : \mathbb{F}_2^n \to \mathbb{F}_2$ of the form:

$$f(x, y) = \langle x, \varphi(y) \rangle, \quad \forall (x, y) \in \mathbb{F}_2^r \times \mathbb{F}_2^{n-r},$$

where $\varphi : \mathbb{F}_2^{n-r} \to \mathbb{F}_2^r$ is an arbitrary function. Here, for $a, b \in \mathbb{F}_2^r$, $a \cdot b$ denotes the standard inner product over $\mathbb{F}_2^r$. In this work, we focus on *signed* variants of Maiorana–McFarland functions, that is, functions of the form

$$g(x) = (-1)^{f(x)} \quad \text{for some } f \in \mathrm{MM}_{r,n}.$$

We now describe the spectral structure of these functions.

**Lemma 4.1** (Proof in the full version [12]). *Let $n = r + \log r$, and suppose $\varphi$ is a mapping whose image has cardinality $r$ and whose elements are linearly independent in $\mathbb{F}_2^r$. Let*

$$g_L(x, y) = (-1)^{\langle Lx, \varphi(y) \rangle}$$

*be a function, where $L \in \mathbb{F}_2^{r \times r}$ is a linear transformation. Then, the Fourier sparsity of $g_L$ is at most* $\mathrm{rank}(L) \cdot r$.

## 4.2 Proof of Theorem 1.3

We prove the lower bound in Theorem 1.3 via a reduction from a variant of the *Approximate Matrix Rank* problem in randomized communication complexity. In this problem, Alice and Bob each hold a matrix in $\mathbb{F}_2^{r \times r}$, denoted by $A$ and $B$, respectively. They are promised that the matrix $C = A + B$ has rank either $r$ or $\frac{r}{4}$, and their task is to determine the correct case while minimizing communication. Both parties have access to a public random string.

Suppose, for the sake of contradiction, that there exists a tester $\mathbb{T}$ which, for any function $f : \mathbb{F}_2^n \to \{-1, 1\}$, distinguishes whether $f$ is $s$-Fourier sparse or $\epsilon$-far from every such function using only $q(s, \epsilon)$ queries. We show that such a tester can be used to solve the matrix rank problem with low communication.

Alice and Bob independently construct Boolean functions $g_A, g_B : \mathbb{F}_2^n \to \{-1, +1\}$ from their matrices $A$ and $B$ using the construction in Lemma 4.1. Define the target function as $g_C : \mathbb{F}_2^n \to \{-1, 1\}$ in similar way. By Lemma 4.1, the Fourier sparsity of $g_C$ depends on the rank of $C = A + B$. Specifically, if $\mathrm{rank}(C) = r$, then the Fourier sparsity of $g_C$ is exactly $r^2$. If $\mathrm{rank}(C) = \frac{r}{4}$, the Fourier sparsity is at most $\frac{r^2}{4}$. Now we show that, in the full-rank case, $g_C$ is $\frac{1}{4}$-far from any Boolean function with Fourier sparsity at most $\frac{r^2}{4}$.

**Lemma 4.2** (Proof in the full version []). *If the matrix $C \in \mathbb{F}_2^{r \times r}$ has rank $r$, then the function $g_C$ defined in Corollary 4.1 is at least $\frac{1}{4}$-far from any Boolean function with Fourier sparsity at most $\frac{r^2}{4}$.*

To simulate the tester $\mathbb{T}$ for $g_C$, Alice and Bob evaluate any query $(x, y) \in \mathbb{F}_2^n$ as follows: Alice computes $g_A(x, y)$, Bob computes $g_B(x, y)$, and they exchange their values. They then compute $g_C(x, y) = g_A(x, y) \cdot g_B(x, y)$. Since

$$g_C(x, y) = (-1)^{\langle (A+B)x, \varphi(y) \rangle} = (-1)^{\langle Ax, \varphi(y) \rangle} \cdot (-1)^{\langle Bx, \varphi(y) \rangle} = g_A(x, y) \cdot g_B(x, y).$$

Each query requires 2 bits of communication. Consequently, if $\mathbb{T}$ uses $q(s, 1/4)$ queries, Alice and Bob can simulate it using at most $2q(s, 1/4)$ bits of communication. Setting $s = \frac{r^2}{4}$, and recalling that distinguishing whether $\mathrm{rank}(C) = r$ or $\frac{r}{4}$ requires $\Omega(r^2)$ bits of communication (Theorem 1.1 from [26]), we deduce $2q(r^2/4, 1/4) = \Omega(r^2)$. Therefore, $q(r^2/4, 1/4) = \Omega(r^2)$.

Thus, any tester distinguishing $s$-Fourier sparse functions from those $\frac{1}{4}$-far from such functions must make at least $\Omega(s)$ queries, establishing the lower bound.

## 5   Conclusion

An intriguing direction is whether similar dimension-independent testers can be obtained for other bases, such as wavelets. While sparsity testing in general bases has been studied before [5], existing algorithms have query complexity that depends on the ambient dimension, making them unsuitable for very high-dimensional settings. In contrast, our work focuses on testers whose query complexity is completely independent of the dimension. Characterizing the classes of functions and bases that admit such testers remains a fundamental open question.

We also highlight a subtle gap between our upper and lower bounds. The upper bound is proved in the tolerant setting, distinguishing functions that are $\delta$-close to being $s$-Fourier sparse from those that are $(\delta + \epsilon)$-far, whereas the lower bound applies to the non-tolerant setting, where functions are either exactly $s$-Fourier sparse or at least $1/4$-far. Since tolerant testing is strictly stronger, and since our algorithm is non-adaptive while the lower bound holds even for adaptive testers, we match the bounds up to a logarithmic factor. Bridging these gaps more fully is an interesting direction for future work.

Finally, our lower bound currently applies only for $\epsilon = 1/4$. Extending it to arbitrary nonzero $\epsilon$ remains an important open problem.

**Acknowledgement.** Arijit Ghosh acknowledges partial support from the MATRICS grant MTR/2023/001527 and the DST grant TPN-104427. A part of this work was carried out during Manmatha Roy's visit to the Selmer Center for Secure Communication at the University of Bergen, hosted by Lilya Budaghyan and partially supported by the Norwegian Research Council. The authors also thank Swagatam Das and Sourav Chakraborty for valuable discussions during the course of this work.

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
