# OpenReview forum: "Price of Parsimony: Complexity of Fourier Sparsity Testing"
_NeurIPS.cc/2025/Conference — NeurIPS 2025 poster_

### Official Review · Reviewer_r9gD · 2025-06-22

**Clarity:** 3
**Significance:** 3
**Originality:** 2
**Rating:** 4
**Confidence:** 4

**Summary:**

This paper studies the property testing problem of Fourier sparsity: It tests whether a given unknown function $f: \mathbb{F}^n_2 \rightarrow \mathbb{R}$ is close to or far from being $s$-Fourier sparse in the $\ell_2$ distance, using a limited number of queries to $f$. This paper shows a (nearly) matching upper and lower bound on the query complexity. The upper bound is shown by testing the function under the restriction to a random affine subspace. By considering a random affine subspace, the tester induces a partition of the spectrum into cosets (viewed as hash buckets), isolating ''heavy'' Fourier coefficients of the function in expectation. Then with a small number of queries, the tester can approximate the contribution from each coset and thus certify whether only few ''heavy'' buckets exist, i.e. the function is sparse. The lower bound is shown via a reduction from the matrix rank problem in the communication complexity. The authors draw a connection between the matrix rank and the Fourier sparsity via a family of functions called Maiorana–McFarland functions.

**Questions:**

1. Could the authors add an explicit comparison between their proof and that of [Gopalan et al. 2011], clearly highlighting which aspects are novel?

2. The reduction connecting matrix rank to Fourier sparsity via a Maiorana–McFarland function seems to be potentially useful beyond the setting in this paper. Have similar connections been observed or applied previously for other purposes?

**Ethical Concerns:**

["NO or VERY MINOR ethics concerns only"]

**Final Justification:**

In the rebuttal, the authors discuss the novelty and similarity in techniques compared to previous works with more details and highlights their contribution. These resolve my initial concerns. Condition on this discussion being incorporated in the paper, I am willing to change my rating upward.

**Limitations:**

See comments in the **Strengths And Weaknesses** section.

**Quality:**

2

**Strengths And Weaknesses:**

**Strength**:


1. The problem definition, the contribution, and the ideas of the proofs are clearly outlined at the beginning of the paper. Therefore, the main body of the paper feels very easy to follow.

2. Testing the Fourier sparsity of a function via queries feels like a natural and important problem. One can easily imagine its usefulness in many downstream applications in the area of property testing, where one might need to assume the Fourier sparsity of a given function. This paper essentially settles this problem under the $\ell_2$ distance, by showing a matching upper ($\tilde{O}(s/\epsilon)$) and lower bound ($\Omega(s)$) of the query complexity. While an upper bound linear in $s$ has already been shown in [Yaroslavtsev and Zhou. 2020], this paper reduces the dependency on $\epsilon$ from $1/\epsilon^4$ to $1/\epsilon$. On the other hand, the improvement over the lower bound, from $\Omega(\sqrt{s})$ to $\Omega(s)$, nicely complements the upper bound.

**Weakness**:

I am concerned that although the upper-bound proof largely builds on the approach of [Gopalan et al. 2011], this connection is never acknowledged. The only mention of [Gopalan et al. 2011] in the $\textbf{Related Work}$ paragraph is that they study a different (Hamming‐distance) version and achieve poly($s$) queries. In my opinion, it would be appropriate to summarize and compare their proof ideas in the subsequent paragraph $\textbf{Regarding the proof of Theorem 1.1}$.

From my understanding, both papers hash the Fourier spectrum via a random subspace restriction to isolate heavy coefficients into separate buckets, and then use concentration‐of‐measure to argue that if too many coefficients collide, then the test will reject. Concretely, both papers hash the Fourier coeffcients into random cosets. After that, both estimate the weight in each bucket by sampling. Finally, both paper argue by pairwise‐independence and Chebyshev/Chernoff‐type bounds that, if the function were far from sparse, then either too many coefficients would collide in the same bucket, or too much Fourier mass would leak into many buckets—so the tester rejects with high probability.

I understand that solving this problem under the $\ell_2$ distance might require new analytic ingredients. But without an explicit comparison to [Gopalan et al. 2011], it is difficult to tell which parts of the proof are novel. Moreover, statements like
> "Surprisingly, despite being a central and natural affine-invariant property, Fourier sparsity has largely resisted similar progress through this lens" (restricting the function to random low-dimensional affine subspaces and tests the property on these restriction),

> "Crucially, we establish new Fourier analytic results that describe how sparsity behaves under such restrictions".

can give the impression that the proof strategy of this paper is entire original, when in fact it closely parallels prior work.

In general, I think adapting the proof from the Hamming distance case to the $\ell_2$ setting can be valid contribution -- indeed, it leads to an improved  $\tilde{O}(s/\epsilon)$ upper bound, which nearly match the new lower bound. However, given the current presentation of this paper, I cannot recommend acceptance.

---

> ### Author Rebuttal · Authors · 2025-07-31
>
> Reviewer's Question-1: Could the authors add an explicit comparison between their proof and that of [Gopalan et al. 2011], clearly highlighting which aspects are novel?
>
> Our Response: We briefly compare our setting and techniques with the work of Gopalan et al. Their goal is to test whether a Boolean function $f: F_2^n \to \lbrace +1, -1 \rbrace $ is $s$-Fourier sparse or at least $\epsilon$-far from every $s$-sparse Boolean function, under the Hamming distance model. In contrast, our setting involves real-valued functions $f: F_2^n \to
> \mathbb{R}$. We aim to distinguish whether there exists an $s$-Fourier sparse function $g: F_2^n \to \mathbb{R}$
> such that $\mid \mid f - g \mid\mid_2 \leq \delta$, or whether $f$ is $(\delta + \epsilon)$-far from every such function in terms of $\ell_{2}$-distance.
>
> To clarify, the Hamming distance between two Boolean functions $f,g: F_2^n \to  \lbrace +1, -1 \rbrace$ is defined as:
> $\mid\mid f - g \mid\mid_0 := Pr_{x \in F_2^n} [ f(x) \neq g(x) ]$,
> while the Euclidean distance between two real-valued functions $f,g: F_2^n \to \mathbb{R}$ is:
> $\mid\mid f - g \mid\mid_2 := \left( E_{x \in F_2^n}[(f(x) - g(x))^2] \right)^{1/2}$.
>
> Thus, both the problem formulation and the nature of the object of interest in this work differ significantly from the Boolean setting of Gopalan et al. Nevertheless, we acknowledge their work as a foundational milestone in the study of Fourier sparsity testing.
>
> As for techniques, the two works also diverge significantly. Gopalan et al. work with the Fourier coefficients of the original Boolean function directly. Their method involves hashing the Fourier spectrum via coset hashing and then estimating the collective weight of the Fourier coefficients falling into a single bucket. To the best of our understanding, their main contribution is to show that for Fourier-sparse Boolean functions, the magnitude of individual Fourier coefficients cannot be too small. Their tester ultimately reduces the problem to counting large-magnitude coefficients.
>
>
> In contrast, our method analyzes the function restricted to a randomly chosen subspace. We approximately (in Euclidean sense)  recover the Fourier spectrum of this restricted function and infer the Fourier sparsity of the original function from it. We derive a new structural relationship between the Fourier coefficients of the restricted function and those of the original function, and show that under a suitable choice of subspace, the magnitudes of these coefficients closely approximate those of the original function. This relationship plays a central role in our analysis.
> We highlight that Gopalan et al.'s hashing of the Fourier spectrum of the original function to a coset still yields a function defined over the full domain $F_2^n$. As we have already discussed, we restrict the original function to a random subspace of the full domain $F_{2}^{n}$.
>
> Finally, we would like to add that our algorithm is extremely simple and lightweight, and we wanted to complete the writing of the paper within the suggested page limit. Therefore, we could not elaborate on all these aspects in the current version. We will definitely expand on these details and provide a more thorough discussion in the final version of the paper.
>
> Reviewer's Question - 2: The reduction connecting matrix rank to Fourier sparsity via a Maiorana–McFarland function seems to be potentially useful beyond the setting in this paper. Have similar connections been observed or applied previously for other purposes?
>
> Our Response: The reduction connecting matrix rank to Fourier sparsity via a Maiorana–McFarland function appears to be potentially useful beyond the setting of this paper. Similar kinds of reductions have been used in the work of Grigorescu, Wimmer, and Xie. In their paper ``Tight Lower Bounds for Testing Linear Isomorphism", they used similar reductions to establish lower bounds for linear isomorphism testing. However, we emphasize that translating their reduction to our context would only yield a lower bound that is logarithmic in the sparsity parameter. Our main contribution lies in refining this framework by invoking a more general version of the rank estimation problem and constructing a specific class of functions that exhibit a controllable trade-off between Fourier dimension and sparsity. We believe that this reduction technique may be of independent interest in other contexts where such dimensionality versus sparsity trade-offs play a central role.
>
> Finally, we sincerely thank the reviewer once again for their valuable time and constructive feedback. We believe that incorporating these suggestions will significantly improve the presentation of the paper. We have made every effort to address each of the comments and concerns to the best of our ability, and we would be happy to engage further should any additional questions arise. We would be truly grateful if the reviewer would kindly reconsider their score for this submission in light of our responses.

---

> > ### Comment · Reviewer_r9gD · 2025-08-04
> >
> > Thank you for your detailed response and for the comprehensive answer to my question 2.
> >
> > Regarding my question 1 (and the weakness section in my review), indeed, the problem formulation in this work is substantially different from the Hamming distance setting studied by Gopalan et al. However, I remain concerned that the similarity in the **proof technique** is more significant than what is conveyed in the response.
> >
> > I agree that the two approaches differ in an ''operational'' view: Gopalan et al.’s hashing takes the Fourier spectrum of the original function and groups coefficients into cosets, so their hash corresponds to a function over the full domain. In contrast, this paper restricts a function to a random affine subspace and analyzes the Fourier spectrum of the restricted function.
> >
> > However, beyond this distinction in the literal sense, the two approaches are closely related at a structural level, and I believe such a relationship should be made more explicit in the paper: Both approaches induce a partition of the original spectrum into cosets, and then estimate aggregated weight per part to detect non-sparsity.  In particular, the methodology in this paper, ''restricting to a random affine subspace'', implicitly implements a kind of spectrum hashing, as the restriction’s spectrum is a sum over original coefficients in cosets. This perspective is also discussed at multiple places in this paper. And I think this is the reason why, e.g., Lemma 2.1 in this paper is closely analogous to Proposition 3 in [Gopalan et al. 2011]. Following the same perspective, I think the concentration studied in this paper, e.g., Lemma 2.2, can be viewed as a new, analytic version of the same ''heavy-bucket detection'' paradigm as in [Gopalan et al. 2011]. Therefore, it feels like saying ''Gopalan et al.’s hashing... still yields a function over the full domain ....... we restrict the original function to a random subspace'' risks framing the difference as bigger than it is, without acknowledging that the effect of the restriction in this paper is a (analytically nontrivial) instantiation of the same underlying hashing intuition.
> >
> > Finally, I understand the pressure of page limits. But I think a clear comparative discussion with prior work could in fact save space and reduce readers' cognitive load: by pointing to the shared high-level skeleton and then isolating precisely what is new (e.g., the structural control in Lemma 2.2), the paper would  make its contributions sharper while appropriately crediting prior work.

---

> ### Author Response · Authors · 2025-08-04
> **Further clarification on the role of Fourier hashing in our work**
>
> We would like to clarify the nature of our main contributions.
>
> The technique of Fourier hashing first appeared in Feldman et al. (New Results for Learning Noisy Parities and Halfspaces, FOCS 2006) and was subsequently used in property testing by Gopalan et al. (SICOMP 2011). Since then, it has become a standard tool in the study of learning and testing Boolean functions. In our work, we do not present the proof of Lemma 2.1, our variant of the Fourier hashing lemma, as a central contribution. Rather, our main contributions are the proposed algorithm, the structural result Lemma 2.2 which analyzes the behavior of the Fourier spectrum under restriction, and the proof of Theorem 1 which establishes the correctness of our algorithm. These points are clearly stated throughout the paper, including in the abstract.
>
> To elaborate, most prior works that employ Fourier hashing use it to derive structural or analytical insights and do not regard the hashing technique itself as a novel contribution. For example, (even) in Gopalan et al. (SICOMP 2011), the main contributions are Theorems 1 and 6, which establish results on the granularity of Fourier coefficients of sparse functions, rather than Proposition 3, which is their Fourier hashing lemma. Similarly, Yaroslavtsev et al., whose result we improve upon, also use Fourier hashing. However, their key contribution is a concentration bound involving the $\ell_2$-norm of hashed coefficients.
>
> Our approach is different both in technique and perspective. Instead of explicitly defining a hash function, we show that restricting the function to a subspace implicitly induces a hashing. This perspective allows us to derive a concentration bound in terms of the $\ell_1$-norm of bucketed Fourier coefficients, stated in Lemma 2.2. In this sense, Lemma 2.2 serves as our central contribution, similar to how Theorems 1 and 6 serve that role in Gopalan et al. We chose to reprove Fourier hashing to meet the specific needs of our proof of Lemma 2.2, since Proposition 3 from Gopalan et al. does not directly apply to our setting, and also to make the presentation accessible to a broader audience. We are happy to include explicit citations to Feldman et al. and Gopalan et al. if the reviewer feels it would improve clarity. However, we believe that presenting Lemma 2.1 as the main novelty misrepresents the essence of our work.
>
> We also wish to respond to the characterization that
>
> ``the concentration studied in this paper, for example Lemma 2.2, can be viewed as a new, analytic version of the same `heavy-bucket detection' paradigm as in Gopalan et al. 2011 . . . . . .  and the effect of the restriction in this paper is an analytically nontrivial instantiation of the same underlying hashing intuition.''
>
> While this description points to a surface-level similarity, we find it somewhat misleading. Heavy-bucket detection is a common goal in sparsity testing; what differentiates algorithms is how this detection is achieved. Our contribution in Lemma 2.2 shows that restriction to small subspaces enables heavy-bucket detection. This is the key new idea. Future upper-bound improvements will likely continue to rely on some form of heavy-bucket detection, but it would be inaccurate to group all such approaches together simply because they share this general objective.
>
> In summary, our problem setting is different, our upper-bound technique is substantially distinct, and we also provide a quadratically stronger lower bound. This shows that even an algorithm of the simplest nature is nearly optimal.
>
> We hope that our results are assessed based on the contributions we have explicitly stated in the paper. Also, describing our work merely as an extension of the Fourier hashing lemma not only misrepresents our contributions, but also those of earlier works such as Gopalan et al. and Yaroslavtsev et al.
>
> We hope this clarification helps in providing a fair assessment of the contribution of our work.

---

> > ### Comment · Reviewer_r9gD · 2025-08-04
> >
> > Thank you for this detailed clarification. I appreciate the effort to situate Lemma 2.1 historically and to emphasize that your core contributions are the subspace‐restriction algorithm and the new structural Lemma 2.2. This discussion of both novelty and similarity in technique is exactly what I think should appear in the paper. For example, as the Fourier hashing procedure is not framed as the main contribution of Gopalan et al. (SICOMP 2011), *this is made clear by explicitly acknowledging "the high-level idea behind our tester is that of 'hashing' the Fourier coefficients, following Feldman et al. (FOCS 2006)" in their paper.* I believe this is a standard thing to do and would indeed improve the clarity for assessing the contribution of this work.
> >
> > I agree that hashing / heavy-bucket detection is a common goal in sparsity testing. But I don't understand why despite mentioning this perspective, the paper does not cite its prior use especially in closely related Fourier-sparsity testing work.  Also, I do not agree that outlining the shared algorithmic structure "groups all such approaches together in an inaccurate way". On the contrary, as said in your response, this "is not what differentiates algorithms", thus making the shared skeleton explicit helps isolate the precise innovations of your work without diminishing their importance.
> >
> > Condition on this comparative discussion being incorporated in the paper, I am willing to change my rating upward.

---

> ### Author Response · Authors · 2025-08-05
>
> We understand your concern. We will make the following changes in the paper:
>
> 1. In Lemma 2.1, we will explicitly mention that it is essentially a restatement of Proposition 3 from Gopalan et al.
>
> 2. We will add a “Proof Idea” section, where we will outline the general strategy, discuss earlier works (if any), and explain our approach for both results of this paper.
>
> We hope these changes address your concern. Please let us know if there is anything else you would like us to clarify or suggest regarding the presentation.

---

### Official Review · Reviewer_XNx6 · 2025-06-27

**Clarity:** 4
**Significance:** 2
**Originality:** 3
**Rating:** 5
**Confidence:** 4

**Summary:**

The paper focuses on the problem of distance estimation of a function $f$ to the closest function which is $s$-sparse in the Fourier basis. To elaborate:
- The function $f$ maps each $x$ in $\\{\pm 1\\}^n$ to a real number $f(x)$. The algorithm can query the value of $f$ on inputs of its choice.
- A function $f’$ over $\\{\pm 1\\}^n$ has Fourier sparsity $s$ if it equals to a linear combination of $s$ functions of the form $\prod_{i \in S} x_i$. For example, the function $f’(x)=x_1+0.1\cdot x_2x_3x_4+5 \cdot x_2x_3$ has Fourier sparsity $3$.
- When speaking of a distance between a pair of functions, the paper considers the $L_2^2$ distance between a pair of functions $f$ and $f’$, which is defined as the average of $(f(x)-f’(x))^2$ over all $x$ in $\\{\pm 1\\}^n$.
- The goal of the algorithm is to approximate, up to an additive error $\epsilon$, the smallest distance of the function $f$ to an $s$-sparse function $f’$.

Motivation for this work: in many settings in machine learning data is approximately sparse in some basis, such as the Fourier basis. This type of tester would allow one to very quickly check whether the data is indeed Fourier-sparse without having to compute all the Foureir coefficients.

This work shows how to accomplish this task using $\tilde{O}(s/\epsilon)$ queries. The work also gives a hardness resulting showing that $\Omega(s)$ queries are necessary.

This and closely related problems have been the focus of multiple previous papers. The most relevant one is [Yaroslavtsev and Zhou 2020] which studied the exact same problem studied here and gave an algorithm that uses $\tilde{O}(s/\epsilon^4)$ queries. [Yaroslavtsev and Zhou 2020]  also showed that $\Omega(\sqrt{s})$ queries are necessary. Thus, when compared with [Yaroslavtsev and Zhou 2020], this work:
- improves the $\epsilon$-dependence for number of queries used by the algorithm from $\tilde{O}(s/\epsilon^4)$ to $\tilde{O}(s/\epsilon)$
- improves the hardness result from $\Omega(\sqrt{s})$ query lower bound to $\Omega(s)$.

The algorithm in this work works by considering a restriction of the function $f$ to a random small-dimensional affine subspace of $\\{\pm 1\\}^n$. The algorithm estimates the distance of the resulting restriction to the closest s-sparse function. It is shown via Fourier-analytic argument that, with high probability, the resulting estimate will be a good approximation to the distance of the original function $f$ to the closest s-sparse function.

The hardness result is shown by a reduction to a problem in communication complexity. In the communication problem, two parties have a matrices A and B respectively, and the goal of the two parties is to approximate the rank of the matrix A+B. It is shown that a $o(s)$-query algorithm for the original task would imply a low-communication protocol for this task, contradicting known communication lower bounds.

**Questions:**

- Judging by the citation record of [Yaroslavtsev and Zhou 2020], their algorithm has not found wide application. Do you think this was only due to the $\epsilon$-dependence of their query complexity, or are there other reasons?

- What is the run-time of the algorithm? Theorem 1.1 only lists the query complexity. How does the run-time compare to that of [Yaroslavtsev and Zhou 2020]?

- In the introduction, you mention sparsity in other bases such as general Fourier basis, wavelet basis and learned dictionaries. What are some obstacles for extending your algorithm to work in these more general settings?

**Ethical Concerns:**

["NO or VERY MINOR ethics concerns only"]

**Final Justification:**

Raising my score to 5 (Accept), based on clarifications given in the rebuttal regarding run-time of the algorithm and potential extensions to wider settings.

**Limitations:**

Yes. Answers to the questions above could potentially help the readers further contextualize some of the limitations of this work.

**Quality:**

4

**Strengths And Weaknesses:**

Strengths

- The problem is quite natural and was studied in multiple papers prior to this work.

- This work gives optimal query bounds up to logarithmic factors.

- The algorithm is quite simple and natural.

Weaknesses

- One could argue that the algorithmic improvement – from $\tilde{O}(s/\epsilon^4)$ to $\tilde{O}(s/\epsilon)$ – over previous work is quite incremental.

Misc comment:

- Given that the improvement over previous work is in $\epsilon$-dependence of the query complexity, I think the $\epsilon$-dependence should be included in the abstract.

---

> ### Author Rebuttal · Authors · 2025-07-31
>
> Reviewer's Question-1:  Judging by the citation record of [Yaroslavtsev and Zhou 2020], their algorithm has not found wide application. Do you think this was only due to the dependence of their query complexity, or are there other reasons?
>
> Our Response: Frankly speaking, we believe that the work of Yaroslavtsev and Zhou (2020) may have missed the attention of its intended audience, possibly due in part to its publication in a theoretical computer science venue.
>
> That said, we would like to emphasize the relevance of their work, and our own, in the broader context of computational learning theory and machine learning. Over the past two decades, there has been sustained interest in the sparse Hadamard transform and compressed sensing-based sparse recovery techniques for learning Fourier-sparse Boolean functions, as such models naturally arise in various machine learning applications. Examples include learning cut functions of graphs and hypergraphs (closely related to hypergraph sketching), as well as evaluating decision trees of bounded depth.
>
> To elaborate, the cut function of a graph corresponds to a polynomial of degree at most $2$ in the Fourier basis, while the cut function of a degree-$d$ hypergraph corresponds to a polynomial of degree at most $d$ (see Stobbe and Krause, 2012). Likewise, a Boolean decision tree of depth $d$ has its Fourier spectrum supported on coefficients of degree at most $d$ (see Mansour, 1994). These functions are therefore naturally Fourier sparse, meaning their Fourier mass is concentrated on low-degree terms. More recently, such Fourier-sparse models have also appeared in applications like neural network hyperparameter optimization (see Hazan et al., 2018). This has led to increased interest in the learning of Fourier-sparse Boolean functions, often referred to as Fourier-sparse set functions (see Amrollahi et al., 2019).
>
> However, both sparse Hadamard transform and compressed sensing-based recovery methods typically require prior knowledge of the sparsity level (and in some cases, the approximation accuracy in the Euclidean setting) to properly configure algorithmic parameters. We strongly believe that our sparsity tester can serve as a valuable preprocessing step in such contexts, by providing reliable estimates of the Fourier sparsity level before applying these recovery techniques.
>
>
> Reference
>
> 1. Indyk, P., Kapralov, M., \& Price, E.: (Nearly) sample-optimal sparse Fourier transform. SODA, 2014.
>
> 2. Kushilevitz, E., \& Mansour, Y.: Learning decision trees using the Fourier spectrum. SIAM Journal on Computing, 1993.
>
> 3. Mansour, Y. (1994). Learning Boolean functions via the Fourier transform. In Theoretical Advances in Neural Computation and Learning, 391–424. Springer.
>
> 4. Rudelson, M., \& Vershynin, R.: On sparse reconstruction from Fourier and Gaussian measurements. Communications on Pure and Applied Mathematics, 2008.
>
> 5. Scheibler, R., Haghighatshoar, S., \& Vetterli, M.: A fast Hadamard transform for signals with sublinear sparsity in the transform domain. IEEE IT, 2015.
>
> 6. Stobbe, P., \& Krause, A.: Learning Fourier sparse set functions. AISTATS, 2012.
>
> 7. Hassanieh, H., Indyk, P., Katabi, D., \& Price, E.: Nearly optimal sparse Fourier transform. STOC, 2012.
>
> 8. Hazan, E., Klivans, A., \& Yuan, Y.: Hyperparameter optimization: a spectral approach. ICLR, 2018.
>
>
> Reviewer's Question-2: What is the run-time of the algorithm? Theorem 1.1 only lists the query complexity. How does the run-time compare to that of [Yaroslavtsev and Zhou 2020]?
>
> Our Response: We regret overlooking this point in our original submission. In fact, our algorithm also improves upon the runtime of [Yaroslavtsev and Zhou, 2020], which is $\tilde{O}(s/\epsilon^4)$, whereas our runtime is $\tilde{O}(s/\epsilon)$. We are truly grateful to the reviewer for bringing this to our attention.
>
> Reviewer's Question - 3: In the introduction, you mention sparsity in other bases such as general Fourier basis, wavelet basis and learned dictionaries. What are some obstacles for extending your algorithm to work in these more general settings?
>
> Our Response: We thank the reviewer for raising this important and insightful question. Our goal is to design sparsity testing algorithms whose query complexity depends only on the sparsity parameter $s$ and the approximation parameter $\epsilon$, and is independent of the ambient dimension $n$. We demonstrate that such dimension-independent testing is indeed possible for functions over $F_2^n$ with bounded $\ell_2$ norm. Functions of this kind arise naturally in machine learning applications, including junta learning, decision tree learning, and hypergraph sketching. We believe the techniques introduced in our work could be extended to other domains where Fourier analysis is applicable.
>
> In particular, our methods naturally generalize to functions $f : F_q^n \to \mathbb{R}$ with bounded $\ell_2$ norm, incurring only a mild $\log q$ multiplicative overhead in complexity. We have chosen to focus on the $F_2^n$ setting in our presentation because such functions are more commonly studied in the computational learning theory literature, and this setting allows our central ideas—particularly the simplicity and near-optimality of our algorithm—to be presented more clearly.
>
> We also believe that similar testing algorithms may be extendable to more general settings, such as functions $f : G \to \mathbb{C}$, where $G$ is a finite Abelian group and Fourier analysis still applies. However, many of our techniques do not directly carry over to this setting. Notably, the notion of a subspace—which is central to our analysis—does not have a straightforward analogue in general finite Abelian groups. Extending our framework would likely require using suitable structural surrogates, such as Bohr neighborhoods. We view this as an exciting direction and are actively exploring it as part of our ongoing research.
>
> We would also like to reiterate that sparsity testing in more general bases has been considered in prior work. In particular, Barman et al. (2018) study related problems where the goal is to test whether a vector (or a collection of vectors) can be expressed as sparse linear combinations over known or unknown bases, using random projections and tools from high-dimensional geometry. However, the query complexity of their algorithms depends on the ambient dimension, making them less suitable for very high-dimensional regimes.
>
> In contrast, our focus is on testing problems where the query complexity is completely independent of the ambient dimension. Understanding precisely which classes of functions and bases allow for such dimension-independent testing is an intriguing and fundamental question.
>
> Reference
>
> 1. Barman, S., Bhattacharyya, A., and Ghoshal, S.: Testing sparsity over known and unknown bases. ICML, 2018.
>
> Finally, we thank the reviewer for their careful reading of our work, for identifying gaps in the presentation, and for suggesting insightful directions for further research. We believe that incorporating these thoughtful suggestions will significantly improve the clarity and overall quality of the paper. We would be happy to address any additional comments or concerns the reviewer may have. We would be truly grateful if the reviewer would kindly consider revising their score in light of our responses.

---

> ### Author Response · Authors · 2025-08-05
>
> We have not heard from you since we posted our rebuttal. We have done our best to address your questions and would be glad to provide any further clarifications on our responses or on any new questions you may have. If you are satisfied with our responses, we would kindly request that you consider increasing the score against our work.

---

### Official Review · Reviewer_tqTn · 2025-06-29

**Clarity:** 4
**Significance:** 3
**Originality:** 3
**Rating:** 5
**Confidence:** 3

**Summary:**

This paper studies the problem of testing if a boolean function is close to an $s$-sparse function (in the fourier domain) or far from it.
They study a simple algorithm which checks if the restriction of the target function $f$ to a random affine subspace is close to being $s$-sparse or far from it. This is a simple non adaptive algorithm that succeeds at testing with probability $2/3$ using $\tilde O(s)$ queries.

They also demonstrate a lower bound of $\Omega(s)$ queries, showing their algorithm is tight up to log factors.

This work improves upon prior work on Boolean functions which solves the same problem with a significantly larger number of queries. Prior work also had a lower bounds of $\Omega(\sqrt{s})$ on the problem, which is quadratically worse than the results of this paper.

I view their main contribution as providing new structural lemmas that describe how Fourier-sparsity operates under restrictions to random subspaces, leading to an improved analysis of a simple algorithm.

**Questions:**

1. Can these techniques be extended to fields beyond $F_2$?
2. For the case of Boolean functions, what is the main reason this algorithm and analysis overcomes prior work?
In this setting aren’t the Hamming and squared 2-norm distances related?
3. This paper required oracle access, is there any reason to expect that random access would result in much worse guarantees? I.e. is there a stronger lower bounds assuming the access to the function is different?

**Ethical Concerns:**

["NO or VERY MINOR ethics concerns only"]

**Final Justification:**

I think this is an interesting paper and vote to accept it.

**Limitations:**

Yes

**Quality:**

3

**Strengths And Weaknesses:**

Strength:
1. The algorithm is quite simple and the analysis is pretty concise.
2. It improves significantly upon the current state of the art to essentially near-optimal.

I think this paper is quite nice and uses a small insight to demonstrate that a simple algorithm is near-optimal.

Weakness:
1. This paper only focuses on the case where the underlying field is $F_2$.
2. I think since the algorithm is extremely simple, it would be good to spend some more time explaining what prior work did and what exactly the main difference is that leads to them having much worse guarantees and this paper having near-optimal guarantees. I think the authors could have perhaps spent some time explaining this contribution a little better.

---

> ### Author Rebuttal · Authors · 2025-07-31
>
> Reviewer's Question - 1: Can these techniques be extended to fields beyond $F_{2}^{n}$
>
> Our Response: We would like to emphasize that our technique naturally extends to functions $f : F_q^n \to
> \mathbb{R}$ with bounded $\ell_2$ norm, with only a mild increase in complexity, specifically, up to a multiplicative $\log q$ factor. We chose to present our analysis primarily for functions over $F_2^n$ since such functions are more commonly studied in computational learning theory. Moreover, to better highlight our central idea—that a simple algorithm can yield near-optimal bounds.
>
> We also believe that similar testing algorithms may extend to the most general setting of Fourier analysis on finite Abelian groups $G$, with functions $f : G \to \mathbb{C}$. However, many of the techniques developed in our work do not extend directly to this broader setting. For instance, the notion of a subspace, which plays a key role in our analysis, does not naturally generalize to arbitrary Abelian groups. One would need to identify suitable structural analogues, such as Bohr neighborhoods, to overcome this challenge. We are actively exploring this direction as part of our ongoing work.
>
> Reviewer's Question - 2: For the case of Boolean functions, what is the main reason this algorithm and analysis overcomes prior work? In this setting aren’t the Hamming and squared 2-norm distances related?
>
> Our Response: First, we emphasize that our setting involves real-valued functions of the form $f : F_2^n \to \mathbb{R}$ with bounded Euclidean norm, which is the most commonly considered setting in machine learning. Functions of the form $g : F_2^n \to F_2$ form a special subset of this broader class. The reviewer is indeed correct in noting that for Boolean-valued functions, Euclidean distance and Hamming distance are equivalent up to a constant factor. However, for real-valued functions, such as the ones we study, the Hamming distance is not a meaningful notion. In the computational learning theory literature, particularly in problems involving the sparse Hadamard transform and sparse recovery, approximation is typically studied in terms of Euclidean ($\ell_2$) distance. This motivates our focus on functions $f : F_2^n \to \mathbb{R}$ and analysis in the Euclidean setting.
>
> We believe that a key technical contribution of our work is the analysis of the Fourier structure of the restriction of a function to a small subspace, which underpins the efficiency of our algorithm. Specifically, we show that when the goal is to estimate the total squared weight of the top $s$ Fourier coefficients, it suffices to analyze the restriction of the function to a suitably chosen low-dimensional subspace. Such low-dimensional projections are a standard technique in the sublinear algorithms literature, particularly in numerical linear algebra, where problems such as matrix-vector multiplication or eigenvalue approximation are addressed by constructing low-dimensional sketches and operating on these reduced representations.
>
> Beyond this projection-based approximation, our work introduces a second important ingredient: a refined analysis of the concentration behavior of the restricted Fourier coefficients. In particular, we derive a novel $\ell_1$ concentration bound for the "bucketed" Fourier coefficients obtained through a Fourier hashing technique. While previous work has primarily focused on $\ell_2$ concentration of such "bucketed" coefficients, typically via techniques analogous to the Count-Min Sketch, our analysis is inspired by the Count Sketch method used for heavy hitter detection in data streams. We believe that this refined concentration analysis is crucial in giving us our near-optimal and conceptually simple algorithm.
>
> Reviewer's Question-3: This paper required oracle access, is there any reason to expect that random access would result in much worse guarantees? I.e. is there a stronger lower bounds assuming the access to the function is different?
>
> Our Response: Testing in random example models is significantly more challenging. To the best of our knowledge, even for the case of linearity testing, where the target functions are $1$-Fourier sparse, it is not known how to design a tester with sample complexity independent of the ambient dimension $n$. In contrast, in the query access model, linearity testing can be performed efficiently using the well-known $3$-query BLR test. The main goal of our work is to estimate the Fourier sparsity level of a function with the number of queries depending only on the sparsity parameter and the proximity parameter, while remaining independent of the ambient dimension $n$.
>
> Finally, we would like to sincerely thank the reviewer once again for their thoughtful and encouraging comments. We appreciate the time and care taken in reviewing our work. We remain very happy to address any further questions or suggestions the reviewer may have.

---

> > ### Comment · Reviewer_tqTn · 2025-08-03
> >
> > Thank you for your detailed and helpful clarification! I will maintain my score.

---

> > ### Author Response · Authors · 2025-08-05
> >
> > We would like to once again thank the reviewer for their thoughtful and engaging feedback.

---

### Official Review · Reviewer_S1nq · 2025-07-01

**Clarity:** 3
**Significance:** 4
**Originality:** 4
**Rating:** 5
**Confidence:** 4

**Summary:**

This paper studies the tolerant property testing question of distinguishing functions over $\mathbb{F}_2^n$ whose Fourier representations are close to being $s$-sparse, from those that are far from any such function in $\ell_2$-distance, using black-box query access to the function. Such a function is $s$-sparse if it has at most $s$ non-zero Fourier coefficients. The goal is to minimize the number of queries. Previously, the state of the art bounds for this question were $\widetilde{O}(s/\epsilon^4)$ and $\Omega(\sqrt{s})$. This paper improves on both fronts, obtaining nearly tight bounds of $\widetilde{O}(s/\epsilon)$ and $\Omega(s)$, thus essentially closing the book on this very natural problem.

Their algorithm works by testing sparsity of the function restricted to a random low-dimensional subspace. Their analysis relies on new Fourier-analytic results regarding the behavior of sparseness under random restrictions.

Their lower bound uses a reduction to the communication complexity problem of determining the rank of the sum of two matrices A,B held by two parties.  Their proof relies on a recent result on this problem due to Sherstov-Storozhenko from FOCS 2024.

**Questions:**

Main questions:

In the proof of Theorem 1.1, you say $\zeta$ is an “estimate” of the distance of the restriction $f_A$ to being $s$-sparse (line 226-227), but then immediately in the next equation you seem to be defining it to be the exact distance (not an estimate). My confusion is also present in line (2) of the algorithm. Is $\zeta$ an estimate or the exact distance? In either case, how are you computing $\zeta$? This is very unclear to me. I also see how in the last paragraph of section 2 you mention that the proof assumes exact values for the Fourier coefficients of the restriction, but approximations suffice which can be computed using the appropriate number of queries. Even though this may be a simple technicality, I think it's important to have a complete rigorous proof by including these details.

Minor typos:

Typo in line 265: “where $f \in MM_{r,n}$ function”

Typo in line 269: “atmost”

**Ethical Concerns:**

["NO or VERY MINOR ethics concerns only"]

**Final Justification:**

This paper obtains a near-optimal algorithm for a very natural testing question. I also think the paper is well-written overall and the authors have addressed the concerns I had raised. Therefore, I think this paper is a clear accept and I am keeping my score a 5.

**Limitations:**

yes

**Quality:**

4

**Strengths And Weaknesses:**

Strengths: This paper obtains essentially tight bounds for what I consider a very natural problem in property testing, which may have nice applications for learning. The proofs are also elegant, and make use of sophisticated results in another area (communication complexity) in a nicely presented, modular way. I think the paper is also well organized and nicely written.

I don’t see any real weaknesses with this work. I do have a question below, which I imagine the authors will be able to address. Conditioned on a satisfactory answer, I believe this paper should absolutely be accepted.

---

> ### Author Rebuttal · Authors · 2025-07-31
>
> Reviewer's Question - 1: In the proof of Theorem 1.1, you say is an “estimate” of the distance of the restriction to being sparse (line 226-227), but then immediately in the next equation you seem to be defining it to be the exact distance (not an estimate). My confusion is also present in line (2) of the algorithm. Is an estimate or the exact distance? In either case, how are you computing? This is very unclear to me. I also see how in the last paragraph of section 2 you mention that the proof assumes exact values for the Fourier coefficients of the restriction, but approximations suffice which can be computed using the appropriate number of queries. Even though this may be a simple technicality, I think it's important to have a complete rigorous proof by including these details.
>
> Our Response: We are really thankful to the reviewer for pointing out the inconsistency. Indeed, the word ``estimate'' in line 226 is incorrect; it should refer to the exact distance, as the analysis in this part assumes access to exact Fourier coefficients. However, the algorithm, as presented in Algorithm~1, uses approximate values of the Fourier coefficients of the restricted function $f_A$. This transition is enabled in the later part of the analysis, in the last paragraph, as pointed out by the reviewer. Below, we provide a detailed explanation which we hope will clarify the apparent gap in the proof.
>
> Since the restriction $f_A$ is over an affine subspace $A \subseteq F_2^n$ of dimension $k = dim(H) = \log \frac{4096s^2}{\epsilon^2}$, one can compute the Fourier coefficients of $f_A$ exactly by querying all points in $A$, which leads to a query complexity of $\widetilde{O}(s^2/\epsilon)$. However, for query efficiency, the algorithm estimates the Fourier coefficients up to the desired accuracy $o(\sqrt{\epsilon/s})$. For each $\alpha \in F_2^k$, the Fourier coefficient is given by $\widehat{f_A}(\alpha) = E_{x \in A} \left[ f_A(x) \cdot \chi_\alpha(x) \right]$, where $\chi_\alpha(x) = (-1)^{\langle \alpha, x \rangle}$. We estimate this by taking $m$ uniform random samples $x^{(1)}, \dots, x^{(m)}$ from $A$ and computing the empirical average: $\widetilde{f_A}(\alpha) := \frac{1}{m} \sum_{i=1}^m f_A(x^{(i)}) \cdot \chi_\alpha(x^{(i)})$. This is an unbiased estimator. Since each term lies in $\lbrace -1, 1\rbrace$, we can apply Hoeffding's inequality to bound the error:
> $Pr\left[ \left| \widetilde{f_A}(\alpha) - \widehat{f_A}(\alpha) \right| > \eta \right] \leq 2\exp(-2m\eta^2)$.
> To ensure additive error at most $\eta = \sqrt{\epsilon/64s}$ with confidence at least $1 - \delta = 1 - \frac{\epsilon^2}{2^{16}s^2}$, it suffices to take
> $m = O\left( \frac{1}{\eta^2} \cdot \log \frac{1}{\delta} \right) = O\left( \frac{s}{\epsilon} \cdot \log  \frac{s}{\epsilon}  \right)$. Applying a union bound over the top $s$ coefficients ensures that all of them are estimated within the desired error with high probability. The total squared error across these coefficients is at most $s \cdot \left( \frac{\epsilon}{64s} \right) = \frac{\epsilon}{64}$, which suffices for the algorithm's decision criterion.
>
> Therefore, the overall query complexity is $\widetilde{O}(s/\epsilon)$.
>
> We thank the reviewer once again for pointing out this inconsistency. We will revise the final version to include the relevant details and ensure that the distinction between the exact analysis and the approximate implementation is made fully rigorous and transparent.
>
> We appreciate all the reviewer’s editorial suggestions/questions and assure them that we will carefully address and incorporate all such comments in the final version of the paper. We truly appreciate the time and care taken in reviewing our work. We believe that incorporating these suggestions will significantly enhance the presentation of the technical content. We remain very happy to address any further questions or suggestions the reviewer may have.

---

> > ### Comment · Reviewer_S1nq · 2025-08-01
> > **Rebuttal response**
> >
> > Thank you for clarifying my confusion regarding the issue of computing/estimating the Fourier coefficients. Your argument about obtaining a sufficient estimate using $\widetilde{O}(s/\epsilon)$ queries makes sense to me. I also appreciate your approach of first presenting the proof assuming you have access to exact values and then dealing with the approximation later. I just think it needs to be very clear to the reader up-front that this is what you're doing and the technical details of the approximation do need to be present (maybe in the appendix) and clearly pointed to where necessary.
> >
> > Again, I think this is a very nice result! Thank you for addressing my questions/concerns. I have no further questions and still support acceptance of the paper.

---

> > > ### Author Response · Authors · 2025-08-05
> > >
> > > We would like to once again thank the reviewer for their thoughtful and engaging feedback. We will incorporate all the suggestions regarding the presentation, and we truly believe these changes will substantially enhance the overall presentation and clarity of the work.

---

### Official Review · Reviewer_qkU2 · 2025-07-03

**Clarity:** 3
**Significance:** 2
**Originality:** 3
**Rating:** 5
**Confidence:** 3

**Summary:**

This paper studies testing the sparsity of the Fourier coefficients of real-valued functions over $F_2^n$. The authors show that $\tilde{O}(s)$ queries to the function is necessary and sufficient to determine whether the Fourier coefficients are approximately $s$-sparse.

**Questions:**

1. It is interesting that the query complexity does not depend on $\delta$. Could the authors provide an intuitive explanation?
2. What is the main difficulty in proving an adaptive lower bound?
3. It has been shown for other problems that under sparsity assumptions, adaptivity could help to achieve better complexity. Could adaptive algorithms close the logarithmic gap?

**Ethical Concerns:**

["NO or VERY MINOR ethics concerns only"]

**Final Justification:**

This paper makes progress in testing sparsity of Fourier coefficientsof boolean functions, which is a fundamental problem in theoretical computer science. The authors addressed my concerns about the lower bound for adaptive algorithms and applications of boolean functions in machine learning. Therefore, my final evaluation is acceptance.

**Limitations:**

The limitations can be found in the theorem statements. Since it is a theoretical paper, there are no immediate social concerns.

**Quality:**

3

**Strengths And Weaknesses:**

Strengths:

1. Fourier coefficients of functions and property testing are very fundamental problems in theoretical computer science. The authors make important progress in these problems by closing the gap for testing sparsity of Fourier coefficients using non-adaptive algorithms.
2. The paper is clearly written and easy to follow.

Weakness:
1. The author does not prove a lower bound with respect to $\epsilon$. Also, from the end of section 1 it seems that the lower bound only holds for non-adaptive algorithms. Therefore, there is still a gap in the results.
2. My main concern is that functions with real domains are more common in machine learning, which have a very different geometric and algebraic structure than $F_2^n$. It is not immediately clear from the paper how $F_2^n$-domain functions and their Fourier coefficients would be useful in applications such as regression. It would be great if the authors could provide examples of $F_2^n$-domain functions and applications of their Fourier coefficients in machine learning.

Overall, this paper presents strong results for a fundamental theoretical problem; however, the relevance to machine learning is not immediately clear.

---

> ### Author Rebuttal · Authors · 2025-07-31
>
> Reviewer's Question - 1: It is interesting that the query complexity does not depend on $\delta$. Could the authors provide an intuitive explanation?
>
> Our Response: Below, we provide an intuitive explanation of our reasoning regarding the independence of $\delta$ in the query complexity.
>
> The central idea is that our algorithm estimates the squared sum of the top $s$ Fourier coefficients of the given function up to an additive error, such as $\epsilon/8$. This level of approximation is sufficient to distinguish whether the function is $\delta$-close or $(\delta + \epsilon)$-far from being $s$-Fourier sparse in Euclidean distance. To see why, recall that, by Parseval’s identity,
> $\mathbb{E}[(f(x) - g(x))^2] = \sum_\alpha (\hat{f}(\alpha) - \hat{g}(\alpha))^2$, so an additive approximation to the Fourier weight of the top $s$ Fourier coefficients suffices for this decision.
>
> In our tester, this estimation is made possible by a concentration phenomenon of the Fourier coefficients under random restriction of the function. Specifically, in Lemma~2.2, we show that when the function is restricted to a random affine subspace, each Fourier coefficient of the restricted function approximates its counterpart in the original function up to an additive error. Leveraging this, and with our choice of parameters, we show that each of the top $s$ coefficients is preserved within an additive error of $O(\sqrt{\epsilon/s})$, and consequently, their squared sum is preserved up to $O(\epsilon)$.
>
> Finally, we would like to draw the reader’s attention to the fact that the query complexity of our tester is determined by the dimension of the random affine subspace used in the restriction. Notably, this dimension, and hence the overall query complexity, depends only on the approximation parameter $\epsilon$ and the sparsity parameter $s$.
>
> Reviewer's Question - 2: What is the main difficulty in proving an adaptive lower bound?
>
> Our Response: Indeed, proving lower bounds for adaptive algorithms is considerably more challenging than for non-adaptive ones. This is because adaptive algorithms can adjust their subsequent queries based on the outcomes of earlier ones, resulting in a significantly larger and more intricate space of strategies that must be ruled out in any lower bound argument.
>
> However, we would like to emphasize that our work actually establishes an adaptive lower bound. This is achieved via a reduction from a suitable two-party communication complexity problem. Such reductions are well-suited for proving adaptive lower bounds, as communication protocols are inherently adaptive, each message may depend on the prior communication. Consequently, the hardness of the communication problem carries over to our model, even when adaptive query strategies are allowed.
>
> Importantly, our upper bound is achieved by a non-adaptive algorithm, while the lower bound applies even to adaptive algorithms. This represents the best of both worlds: the algorithm operates in a weaker (non-adaptive) setting, yet the lower bound holds in a stronger (adaptive) setting. This is one of those interesting cases where adaptivity offers no advantage over non-adaptivity. We will revise the exposition in the final version to highlight and clarify this more explicitly.
>
> Reviewer's Question - 3: It has been shown for other problems that under sparsity assumptions, adaptivity could help to achieve better complexity. Could adaptive algorithms close the logarithmic gap?
>
> Our Response: In addition to the issue of adaptivity versus non-adaptivity, we would like to draw attention to another subtle aspect of our work. Specifically, our upper bound is established in the tolerant setting, where the goal is to distinguish functions that are $\delta$-close to being $s$-Fourier sparse from those that are $(\delta + \epsilon)$-far from every $s$-Fourier sparse function. In contrast, our lower bound is proved in the non-tolerant setting, where the function is either exactly $s$-Fourier sparse or at least $1/4$-far from every such function. Clearly, the tolerant model is slightly stronger, as it requires accepting all functions that are close to being $s$-Fourier sparse, not just those that are exactly $s$-Fourier sparse. On the other hand, by definition, a non-tolerant tester is not required to behave in any particular way on such close to being $s$-Fourier sparse functions. It is worth noting that in the property testing literature, logarithmic or even larger gaps between tolerant and non-tolerant complexities have been observed in several problems.
>
> Returning to our discussion, we would like to highlight that our contribution consists of a non-adaptive tolerant tester and an adaptive lower bound in the non-tolerant setting.
>
> Frankly speaking, at this point, we do not know which of our bounds are tight. However, we view this as an interesting and open question, and we are actively working to better understand the complexity of Fourier sparsity testing across all possible settings.
>
> Reviewer's Remark - 1: The author does not prove a lower bound with respect to $\epsilon$. Also, from the end of section 1 it seems that the lower bound only holds for non-adaptive algorithms. Therefore, there is still a gap in the results.
>
> Our Response: Our lower bound argument can be adapted to get $\Omega(s/\epsilon)$ lower bound. Note that our lower bound is for adaptive algorithms.
>
> Reviewer's Remark - 2: My main concern is that functions with real domains are more common in machine learning, which have a very different geometric and algebraic structure than $F_2^n$. It is not immediately clear from the paper how $F_2^n$-domain functions and their Fourier coefficients would be useful in applications such as regression. It would be great if the authors could provide examples of $F_2^n$-domain functions and applications of their Fourier coefficients in machine learning.
>
> Our Response: Indeed, functions over real domains are common in many classical machine learning tasks. However, we want to emphasize that the problem of learning Fourier sparse functions defined over the Boolean hypercube $F_2^n$ has been a central theme in computational learning theory and has found important applications across various machine learning settings. Notable examples include learning cut functions of graphs and hypergraphs (which are closely related to hypergraph sketching), as well as evaluating decision trees of bounded depth.
>
> To illustrate further, the cut function of a graph corresponds to a polynomial of degree at most $2$ in the Fourier basis, while the cut function of a degree $d$ hypergraph corresponds to a polynomial of degree at most $d$ (see Stobbe and Krause, 2012). Similarly, a Boolean decision tree of depth $d$ has a Fourier spectrum supported on coefficients of degree at most $d$ (see Mansour, 1994). Hence, these functions are naturally Fourier sparse, i.e., their Fourier spectrum is concentrated on low-degree terms. More recently, Fourier sparse functions have appeared in applications such as neural network hyperparameter optimization (see Hazan et al., 2018). Due to these, there has also been growing interest in learning Fourier sparse Boolean functions, also known as Fourier sparse set functions (see Amrollahi et al., 2019).
>
> In addition to machine learning, learning Fourier sparse Boolean functions also plays a central role in cryptography. For example, the Goldreich-Levin theorem (Goldreich and Levin, 1989) shows how to construct hardcore predicates from any one-way function using a procedure that relies on learning large Fourier coefficients of certain Boolean functions.
>
> Finally, we comment that learning Fourier sparse functions falls into two broad categories of methods: Sparse Hadamard Transform (see Hassanieh et al., 2012; Indyk et al., 2014; Scheibler et al., 2015) and Compressed Sensing techniques (see Rudelson and Vershynin, 2008). However, both these lines of work assume prior knowledge of the sparsity level in the Fourier domain. Our work can thus serve as an important preprocessing step to efficiently estimate this sparsity level, up to desired approximation level in terms of Euclidean distance, which is also the natural distance measure for both of these approaches. We will revise the manuscript to better highlight the relevance of our results to the different areas of machine learning.
>
> Reference:
>
>   1. Amrollahi, A., Zandieh, A., Kapralov, M., & Krause, A.: Efficiently learning Fourier sparse set functions. NeurIPS, 2019.
>
>   2.  Hassanieh, H., Indyk, P., Katabi, D., & Price, E.: Nearly optimal sparse Fourier transform. STOC, 2012.
>
>   3.  Hazan, E., Klivans, A., & Yuan, Y.: Hyperparameter optimization: a spectral approach. ICLR, 2018.
>
>   4.  Indyk, P., Kapralov, M., & Price, E.: (Nearly) sample-optimal sparse Fourier transform. SODA, 2014.
>
>   5. Kushilevitz, E., & Mansour, Y.: Learning decision trees using the Fourier spectrum. SICOMP, 1993.
>
>   6.  Mansour, Y.: Learning Boolean functions via the Fourier transform. 1994
>
>   7.  Rudelson, M., & Vershynin, R.: On sparse reconstruction from Fourier and Gaussian measurements. Commun. on Pure & App. Math., 2008.
>
>   8.  Scheibler, R., Haghighatshoar, S., & Vetterli, M.: A fast Hadamard transform for signals with sublinear sparsity in the transform domain. IEEE IT, 2015.
>
>   9. Stobbe, P., & Krause, A.: Learning Fourier sparse set functions. AISTATS, 2012.
>
>   10. Goldreich, O., &  Levin, L.: A hard-core predicate for all one-way functions. STOC, 1989.
>
>
> We sincerely thank the reviewer for their careful reading of our work and for pointing out gaps in the presentation. We believe that incorporating these thoughtful concerns  will greatly enhance the clarity and overall quality of the paper. We would be glad to address any further comments or concerns the reviewer may have. We would be truly grateful if the reviewer would kindly consider revising their score in light of our responses.

---

> > ### Comment · Reviewer_qkU2 · 2025-08-04
> >
> > Thank you for your detailed response. You have addressed most of my concerns. If no other major concern is found, I am willing to increase my score.

---

> > > ### Author Response · Authors · 2025-08-05
> > >
> > > We would like to once again thank the reviewer for their thoughtful and engaging feedback.
> > >
> > > We will incorporate all the suggestions regarding the presentation, and we truly believe these changes will substantially enhance the overall presentation and clarity of the work.

---

### Note · Authors · 2025-08-13

We once again thank all the reviewers for their helpful comments. We will make the following revisions to the paper:

1. We will include an intuitive explanation of the independence of $\delta$ in the query complexity of our algorithm in Section 2.

2. We will highlight the fact that our algorithm is tolerant and non-adaptive, whereas the lower bound is for adaptive and non-tolerant algorithms. We include further research directions in this regard in a separate (to-be-added) conclusion section.

3. We will include a discussion on the importance of functions of the form $\mathbb{F}_2^n \to \mathbb{R}$ in various areas of machine learning and cryptography, where our work could be particularly useful, in Section 1.

4. We will elaborate more on the transition from exact computation of Fourier coefficients to the approximate computation of Fourier coefficients of the restricted function in the analysis of the Algorithm in Section 2.

5. We will also highlight the fact in Section 1 that the problem is extremely challenging in random example models, where the user does not have query access but only uniform samples of the function.

6. We will discuss the extendability of this work for functions over other domains and the feasibility of determining sparsity in different bases, such as wavelet bases, and learned dictionaries, in the conclusion (to-be-added) section.

7. We will explicitly mention the similarity of  Lemma 2.1 to Proposition 3 from Gopalan et al. Additionally, in the "Proof Idea" section, we will elaborate more on the general strategy, connect it with earlier related works, and explain our approach for both results in this paper.

8. We will modify the lower bound argument to show its dependence on $\epsilon$ in Section 3.

9. We will explicitly state the runtime of the algorithm in the upper bound theorem statement.

We strongly believe that these revisions will significantly improve the presentation of our work and make it suitable for a wider audience, considering its broad applicability in the context of machine learning.

---

### Decision · Program_Chairs · 2025-09-17

**Decision:**

Accept (poster)

**Comment:**

This paper studies testing the sparsity of the Fourier coefficients of real-valued functions over the boolean hypercube. The authors give an algorithm that uses $\tilde{O}(s/\varepsilon)$ queries to determine whether the Fourier coefficients are $s$-sparse or at least $\varepsilon$-far from $s$-sparse in squared $L_2$ distance. They also achieve a lower bound of $\Omega(s)$. By comparison, the previous upper and lower bounds were $\tilde{O}(s/\varepsilon^4)$ and $\Omega(\sqrt{s})$. All reviewers agree that these improvements are quantitatively valuable, the algorithm is quite simple, and the manuscript is generally accessible.

During the rebuttal phase, the authors suggested a number of actionable items in direct response to the initial reviews. I believe these changes could strengthen the overall impact of the work and thus I encourage the authors to take these points into consideration.